# A putative ABC transporter gene, *CcT1*, is involved in beauvericin synthesis, conidiation, and oxidative stress resistance in *Cordyceps chanhua*

Zhimin Liu,[1] Jiahua Zhu,[1,2] Ruixue Gong,[1] Zhiyuan Wen,[1] Yanwen Liu,[1,3] Yulong Wang,[1] Zengzhi Li,[1] Bo Huang,[1] Fan Peng[1]

**ABSTRACT** *Cordyceps chanhua* is a traditional Chinese medicinal fungus renowned for producing a variety of bioactive compounds, including beauvericin (BEA). BEA has garnered significant attention due to its therapeutic potential and associated food safety concerns. In this study, we identified an ATP-binding cassette (ABC) transporter-encoding gene, *CcT1*, located within the BEA synthesis gene cluster of *C. chanhua*. Disruption of *CcT1* resulted in a substantial decrease in BEA production. RT-qPCR analysis demonstrated that the loss of *CcT1* significantly downregulated the expression of several BEA synthesis-related genes, including pyruvate kinase, branched-chain amino acid aminotransferase, and ketoisovalerate reductase. Beyond its role in BEA biosynthesis, *CcT1* was found to influence hyphal growth, conidiation, conidial germination, and the oxidative stress response in *C. chanhua*. Additionally, the *CcT1* knockout strain exhibited a reduced ability to penetrate host cuticles, highlighting the gene's role in fungal pathogenicity. These findings offer a comprehensive understanding of the multifaceted roles of the ABC transporter CcT1 in hyphal development, conidiation, BEA biosynthesis, and stress resistance in *C. chanhua*. Moreover, targeting *CcT1* presents a promising strategy for reducing BEA content through molecular breeding, thereby enhancing the safety and efficacy of *C. chanhua* as a medicinal agent.

**IMPORTANCE** Beauvericin (BEA) is one of the bioactive components in *Cordyceps chanhua*, a significant medicinal fungus with widespread use in Asia and beyond. BEA also possesses mycotoxin properties, with certain cytotoxicity and potential *in vivo* toxicity. However, few studies report the regulation of BEA anabolism. ABC transporters are a superfamily of membrane proteins and have multiple functions such as regulating fungal metabolism. Here, we report an ABC transporter CcT1 involved in BEA synthesis. The disruption of its encoding gene *CcT1* led to a 64.22% reduction in BEA content compared to the wild-type by regulating the expression levels of several BEA synthesis-related genes. It also affected hyphal growth, conidiation, spore germination, penetration, and oxidative stress resistance of the fungus. The findings in this study enrich the understanding of the function of ABC transporter in fungal metabolism and growth and development.

**KEYWORDS** *Cordyceps chanhua*, ABC transporter, beauvericin, vegetative growth, conidiation, oxidative stress resistance

*C*ordyceps chanhua is a fungus with a long-standing history in traditional Chinese medicine, esteemed for its diverse therapeutic properties (1). Modern research has identified that *C. chanhua* synthesizes a variety of bioactive compounds, including nucleosides, polysaccharides, cordycepic acid, multiochrocetin, and beauvericin (BEA), each contributing to its wide-ranging pharmacological activities (2). These activities

**Peer Reviewers** Yongjun Zhang, Southwest University, Chongqing, China; Caihong Dong, Institute of Microbiology Chinese Academy of Sciences, Beijing, China

Address correspondence to Fan Peng, fpeng@ahau.edu.cn.

Zhimin Liu and Jiahua Zhu contributed equally to this article. Author order was determined alphabetically.

The authors declare no conflict of interest.

See the funding table on p. 16.

encompass antitumor, immunomodulatory, neuroprotective, renal function improvement, hypoglycemic, and antibacterial effects (2). Notably, the biological activities and bioactive profiles of *C. chanhua* closely resemble those of *Ophiocordyceps sinensis* and *C. militaris*, positioning it as a viable alternative to these well-studied species (3). This potential has been realized in China, where *C. chanhua* is cultivated on a large scale (4), and its commercial products, such as "Cikaria" available in Sweden, are marketed as health supplements (3).

Among the bioactive compounds produced by *C. chanhua*, BEA has attracted significant attention due to its dual role as a therapeutic agent and a mycotoxin. BEA, a cyclic hexadepsipeptide, was first isolated from the entomopathogenic fungus *Beauveria bassiana* (5) and has been identified in other fungal genera such as *Fusarium* and *Cordyceps* (6). BEA exhibits a range of bioactivities, including antibacterial (7), anti-inflammatory, and anticancer properties (8, 9), making it a promising candidate for drug development. However, BEA is also recognized as a mycotoxin with cytotoxic effects and potential *in vivo* toxicity (10). These dual aspects have led regulatory authorities, including the European Food Safety Authority (EFSA) and Chinese National Food Safety Standards, to impose limits on BEA levels in food products (11). The variability in BEA production is influenced by multiple factors, including fungal strain differences, culture medium composition, and cultivation conditions (12). Despite its importance, the regulatory mechanisms governing BEA biosynthesis remain poorly understood, with current research predominantly focusing on the genera *Fusarium* and *Beauveria*.

The biosynthetic pathway of BEA was first elucidated in *B. bassiana* in 2008, when Xu et al. cloned the BEA synthesis gene cluster, identifying key nonribosomal peptide synthetase (NRPS) genes *bbBeas* and *kivr* (ketoisovalerate reductase encoding gene) essential for BEA production (13). BbBEAS catalyzes the synthesis of the N-methyl-dipeptidol intermediate from D-hydroxyisovalerate (*D-Hiv*) and L-phenylalanine (*Phe*), a precursor to BEA (13). Subsequently, Zhang et al. characterized the *fpBeas* gene cluster in *F. proliferatum*, revealing a more complex and integrative function compared to *bbBeas* (14). These studies underscore the intricate genetic and enzymatic networks involved in BEA biosynthesis.

Building on our previous research (15), which indicated that the BEA content in *C. chanhua* is modulated by oxidative stress, we conducted a comprehensive transcriptomic analysis under oxidative stress conditions. This analysis led to the identification of an ATP-binding cassette (ABC) transporter-encoding gene, *CcT1*, which exhibited significant expression changes correlated with BEA content alterations. Notably, *CcT1* is situated within the BEA synthesis gene cluster of *C. chanhua*. ABC transporters are a highly conserved and versatile family of transmembrane proteins ubiquitous in fungi, playing critical roles in the transport of diverse substrates across cellular membranes through ATP hydrolysis (16, 17). They are integral to fungal metabolism, responsible for exporting metabolites and toxins, maintaining intracellular homeostasis, and enhancing tolerance to drugs and environmental stresses (18–20).

ABC transporters have been implicated in the regulation of secondary metabolite biosynthesis across various fungal species. For instance, in *Leptosphaeria maculans*, disruption of the ABC transporter gene *sirA* within the sirodesmin gene cluster resulted in increased sirodesmin secretion and upregulation of the sirodesmin synthesis gene *sirP* (21). Similarly, in *F. graminearum*, the ABC transporter ZRA1 is associated with zearalenone (ZEA) production (22), and in *Cercospora nicotianae*, ATR1 is linked to cercosporin production (23). In *Aspergillus nidulans*, the ABC transporter atrD facilitates penicillin secretion (24). These examples collectively suggest that ABC transporters are pivotal in the production and regulation of mycotoxins, highlighting their potential role in BEA biosynthesis in *C. chanhua*.

In this study, we aim to elucidate the regulatory role of the ABC transporter CcT1 in BEA biosynthesis and biological function in *C. chanhua*. By comparing the wild-type strain with the *CcT1* knockout mutant, we investigate the impact of *CcT1* disruption on BEA production and related physiological processes. Understanding the multifaceted

roles of *CcT1* not only advances our knowledge of fungal metabolism and pathogenicity but also offers potential strategies for molecular breeding to reduce BEA content, thereby enhancing the safety and efficacy of *C. chanhua* as a medicinal agent.

## MATERIALS AND METHODS

### Fungal strains and culture

In this study, the wild-type strain utilized for transformation is *C. chanhua* RCEF5833, which was isolated from bamboo cicadas *Platylomia pieli* Kato (Hemiptera: Cicadidae) collected in Jingtingshan, Xuancheng, in Anhui province, China, and preserved at the Research Center on Entomogenous Fungi (RCEF). All strains, including the wild-type and mutants, underwent a cultivation period of 10 days on Sabouraud dextrose agar with yeast extracts (SDAY). Following this incubation period, conidia were collected and dispersed by vortex in a 0.05% Tween-80 aqueous solution.

### Sequence analysis

To determine the phylogenetic relationship among *CcT1* and their orthologs, the amino acid sequences of those functionally characterized CcT1 subunits of different entomogenous fungi were downloaded from the National Center for Biotechnology Information (NCBI) database (https://www.ncbi.nlm.nih.gov/). Following this, we conducted protein domain analysis by using the online software SMART (https://smart.embl-heidelberg.de/). Finally, to explore the evolutionary relationships among these fungi, we constructed a phylogenetic tree using the neighbor-joining method in the MEGA 7 software (http://www.megasoftware.net/).

### Subcellular localization of CcT1

To determine the subcellular localization of *CcT1*, *gfp* and *CcT1* gene fragments were amplified by the polymerase chain reaction (PCR), using *gfp-F/gfp-R*, and *gfpCcT1-F/gfp CcT1-R* primers, high-fidelity Taq DNA polymerase (KOD Plus Neo, Toyobo, Osaka, Japan), and *C. chanhua* genomic DNA as a template. The amplification products were inserted into the *EcoRI* restriction site in the pDHt-SK-*bar* vector (kindly provided by Dr. Chengshu Wang; the vector conferred resistance against glufosinate-ammonium) (25) containing a strong promoter and terminator to generate vector pDHt-*CcT1-gfp* for *A. tumefaciens* transformation. The corresponding transformants resistant to glufosinate ammonium were obtained and verified by the PCR using the primers *gfp-F* and *gfp-R*.

The subcellular localization of the CcT1-GFP fusion protein in conidia and mycelia was visualized using confocal laser scanning microscopy (CLSM, Zeiss LSM980, Zeiss Gruppe, Oberkochen, Baden-Wurttemberg, Germany). A positive transformant showing a strong green signal (expressed CcT1-GFP fusion protein) was selected for cultivation on SDAY until full conidiation. Collected conidia were suspended in SDB (agar-free SDAY) and static incubated at 25°C for 2 days. Following the completion of cultivation, conidia and hyphae were subjected to organelle-specific staining protocols. Cell membranes were stained with membrane dye FM4-64 (Coolaber, CD4673) (26) for 10 minutes. Vacuolar compartments were stained with 7-amino-4-chloromethylcoumarin (CMAC; Thermo Fisher Scientific, C2110) (27) for 5 minutes. Confocal laser scanning microscopy (CLSM) is used to analyze and evaluate the overlap between the green fluorescence of fusion proteins and the staining color of specific organelle markers.

### Gene deletion and complementation

Targeted gene disruption of *CcT1* was performed by homologous recombination via *Agrobacterium tumefaciens* transformation, as previously described. Slightly modified from the method of Youmin Tong (28). In brief, the 5′ and 3′ flanking regions of *CcT1* were inserted into the pDHt-SK-*bar*, and then the vectors pDHt-*CcT1-bar* were obtained for fungal transformation.

For mutant complementation, the entire *CcT1* gene was amplified in the upstream and downstream regions and then inserted into the vector pDHt-SK-*ben* to produce the vectors pDHt-C-*CcT1-ben* for fungal transformation. Next, all of the transformants were verified by DNA sequencing. All the primers used in this study are listed in Supplementary materials, Table S1.

## Phenotype assays

For vegetative growth assessments, 1 µL aliquots of the conidial suspension (1 × $10^7$ conidia/mL, the same below unless otherwise specified) of different strains were uniformly spotted onto agar plates with different nutrient compositions, including potato dextrose agar medium (PDA), SDAY, and 1/4 SDAY (containing 1/4 of the nutrients SDAY). The fungal colonies were meticulously documented, and the diameter of each colony was measured. Growth indices were calculated following a 14-day incubation period at 25°C (29).

For germination assessments, 10 µL of the conidial suspension (1 × $10^6$ conidia/mL) was applied at the center of GB plates without spreading. The plates were then incubated for 24 hours, and the percentage of germinated conidia on each plate was determined through microscopic examination at 2 hour intervals, starting from 2 hours. Germination was considered to have occurred when the germ tube lengths were approximately equal to conidia (30). Three hundred conidia were counted at least per plate, and the germination rates were calculated by comparing the number of germinated conidia with 300 counted conidia (31), and the median germination time ($GT_{50}$) was calculated using the SPSS software.

For the assay of conidial yield, 1 µL aliquots of conidial suspensions of different strains were uniformly spotted onto agar plates with different nutrient compositions, including PDA plates (90 mm diameter), SDAY plates, and 1/4 SDAY plates and cultured at 25°C for 7 and 14 days. Subsequently, conidia from each plate were suspended in 50 mL of a 0.05% Tween 80 solution and thoroughly mixed to eliminate any mycelial debris by sterile non-woven fabric filtration. The concentration of conidial suspensions was determined by using a hemocytometer (32).

For evaluating the ability of cuticle penetration, conidial suspensions were applied to the center of intact cicada wings on SDAY plates and cultured at 25°C for 2 or 4 days. Subsequently, the cicada wings were removed, and the SDAY plates were further incubated for 5 days at 25°C. Fungal colony size was measured, and the images were captured.

For assay of chemical stresses, 1 µL of conidial suspensions from different strains was applied to individual SDAY plates (control) or SDAY plates containing the corresponding stress-inducing agents, such as Congo red (200 µg / mL or 600 µg / mL) to disrupt cell wall integrity and menadione (20 µmol / L or 40 µmol / L) and $H_2O_2$ (2 mmol / L or 3 mmol / L) for oxidative stress. After incubation at 25°C for 14 days, fungal colony size was measured, and the relative inhibition rate was calculated (33).

Fungal virulence was assessed by using *Galleria mellonella* larvae. A 10 µL of conidial suspension (1 × $10^5$ conidia/mL) was injected into the hemocoel and incubated at 25°C. Each treatment was performed in triplicate, with 18 larvae in each group. The experiment was repeated three times. Larva mortality was evaluated every 24 hours, and the median lethal time ($LT_{50}$) was calculated using SPSS software.

To assay the formation of synnema, 1 mL conidial suspension (1 × $10^5$ conidia/mL) of the strains was injected into the hemocoel of the pupae of Chinese tussah silkworm (*Antheraea pernyi*). Incubation was performed at 17°C in the dark for 14 days, at 20°C in the light for 12 hours and at 16°C in the dark for 12 hours for 5 days, and at 25°C in the light for 16 hours and in the dark for 8 hours for 14 days. The assay was performed in triplicate, with 20 insects per replicate for each strain (34).

## BEA production assay

The content of BEA was determined as described previously (12). The BEA yield was calculated using the detected peak area according to the standard curve. The BEA concentration of mycelia presented in our study was calculated by normalizing in the equal biomass.

## Quantitative real-time RT-qPCR analysis

To analyze the expression of genes related to BEA synthesis, total RNA was acquired from different strains incubated on SDAY plates for a 6 day duration. The relative expression levels of genes were quantified by using the $2^{-\Delta\Delta Ct}$ method (35). The primer sets required for RT-qPCR analysis are provided in Table S2.

## Statistical analysis

Each experiment was performed using three technical repeats. IBM SPSS Statistics 25 software was used to determine significant differences, and F-Test was used for data analysis. $P < 0.05$ indicated statistical significance.

## RESULTS

### Identification and characterization of CcT1 in *C. chanhua*

In our previous studies, an analysis of transcriptomic data under oxidative stress revealed that the expression levels of some putative ABC transporters were influenced by oxidative stress, with CL3361.Contig1_All showing significantly different expression levels. We utilized SMART and TMHMM online analysis software to predict its secondary structure and transmembrane characteristics. Its structure is presumed to resemble the (TMD-NBD)2 configuration of the ABCB transporter. Consequently, we compared all known full-length ABCB amino acid sequences in entomogenous fungi and constructed a phylogenetic tree. This analysis showed that CL3361.Contig1_All shares the closest homology with the ABCB transporter ISF_00177 from *C. fumosorosea* and shared 94.28% amino acid sequence similarity. Furthermore, this gene is located within the BEA synthetic gene cluster. Given the significant changes in BEA content observed in *C. chanhua* under oxidative stress in our previous study, we speculate that this transporter is related to BEA biosynthesis, leading us to identify it as a focus of our research and designate it as *CcT1*.

This gene is located upstream of the KIVR-encoding gene within the BEA synthesis gene cluster. A schematic map showed the putative BEA gene cluster in different species (Fig. 1). *CcT1* spans 4,643 base pairs and encodes a protein of 1,291 amino acids with a molecular weight of approximately 169.06 kDa. Analysis of conserved domains using the SMART database revealed that the N-terminal region of CcT1 contains two ABC_membrane domains (Fig. 2A). Further prediction of the secondary structure and transmembrane regions using TMHMM software indicated that CcT1 possesses a (TMD-NBD)2 configuration typical of ABCB transporters. Sequence alignment demonstrated that CcT1 shares high homology with the ABC transporter protein ISF_00177 from *C. fumosorosea* and BBA_00005 from *B. bassiana*. Phylogenetic analysis showed that CcT1 is evolutionarily closely related to these transporters in the ABC family (Fig. 2B), suggesting a conserved function in fungal metabolism.

### Generation and verification of *CcT1* knockout and complementation strains

To elucidate the function of *CcT1*, we generated a knockout mutant (Δ*CcT1*) and a complementation strain (C-Δ*CcT1*) via homologous recombination (Fig. S1A). PCR amplification confirmed the successful deletion of *CcT1* in the Δ*CcT1* strain as the 2,040 bp fragment corresponding to the *CcT1* gene was absent. In contrast, the wild-type and C-Δ*CcT1* strains retained the *CcT1* gene fragment. Furthermore, the PCR using primers P3 and P4 verified the presence of the complementation construct in C-Δ*CcT1*,

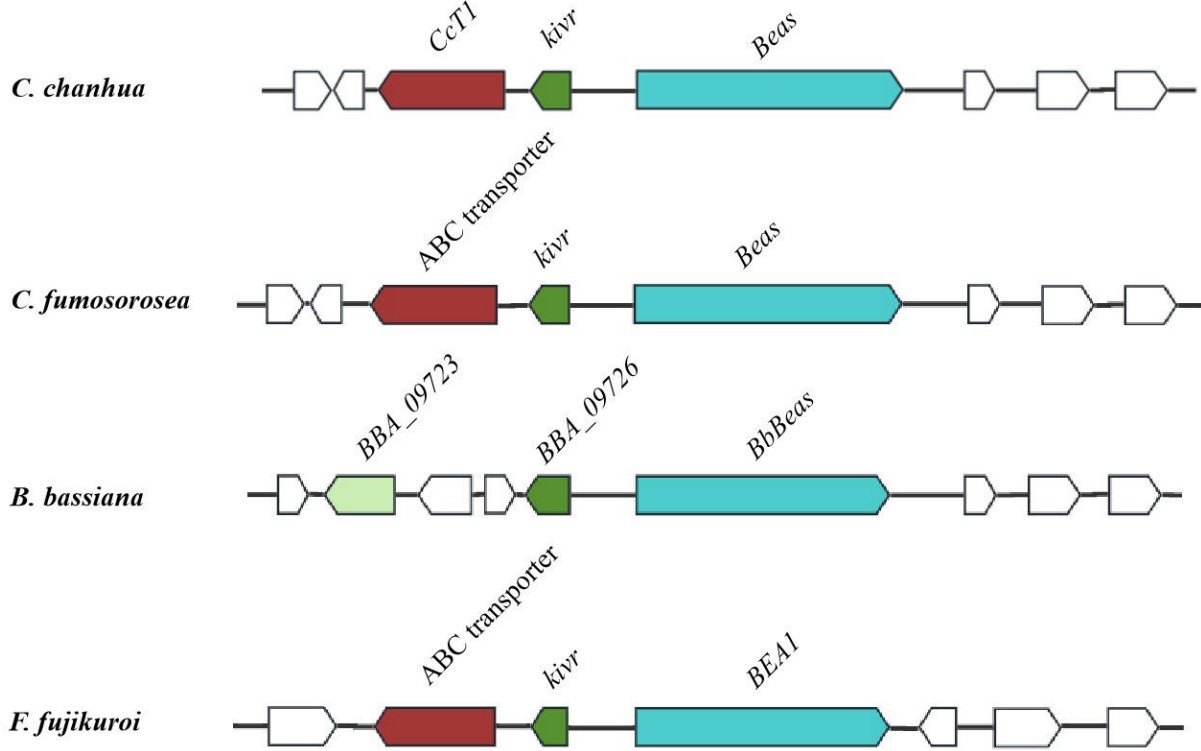

**FIG 1** Schematic map of the beauvericin biosynthetic gene cluster in different fungi (36, 37).

with no amplification observed in ΔCcT1. These results confirm the successful generation of both the knockout and complementation strains.

## Subcellular localization of *CcT1*

To determine the subcellular localization of *CcT1*, we constructed *CcT1*-eGFP fusion vectors and performed fungal transformation. Confocal microscopy revealed significant colocalization between the CMAC-stained vacuolar compartments (Klein blue fluorescence) and *CcT1*-eGFP signals (green fluorescence) in both conidia and hyphae (Fig. 3A). FM4-64 membrane tracking experiments demonstrated precise overlapping of cell membrane signals (red fluorescence) with *CcT1*-eGFP localization, as evidenced by yellow merged signals (Fig. 3B). These dual localization patterns conclusively demonstrate that *CcT1* is targeted to both vacuole and cell membrane systems in *C. chanhua*.

## Impact of *CcT1* Deletion on Vegetative Growth

We assessed the vegetative growth of the WT, ΔCcT1, and C-ΔCcT1 strains on three different media: PDA, SDAY, and 1/4 SDAY under varying nutritional conditions (Fig. 4A). On PDA and 1/4 SDAY, which are relatively nutrient-poor, the ΔCcT1 strain exhibited significantly slower growth compared to the WT and C-ΔCcT1 strains ($P < 0.05$). However, on the nutrient-rich SDAY medium, no significant differences in growth rates were observed among the strains. These findings indicate that *CcT1* plays a critical role in the uptake and utilization of carbon and nitrogen sources, particularly under limited nutrient conditions, thereby influencing vegetative growth.

## Effects of *CcT1* on conidiation and synnema formation

We evaluated conidial germination and production in the WT, ΔCcT1, and C-ΔCcT1 strains. The mean 50% germination time (GT$_{50}$) was significantly reduced in the ΔCcT1 strain (10.46 ± 0.09 hours) compared to the WT (11.95 ± 0.13 hours) and C-ΔCcT1 (11.71 ± 0.10 hours) strains (Fig. 5B), indicating accelerated germination upon *CcT1* deletion.

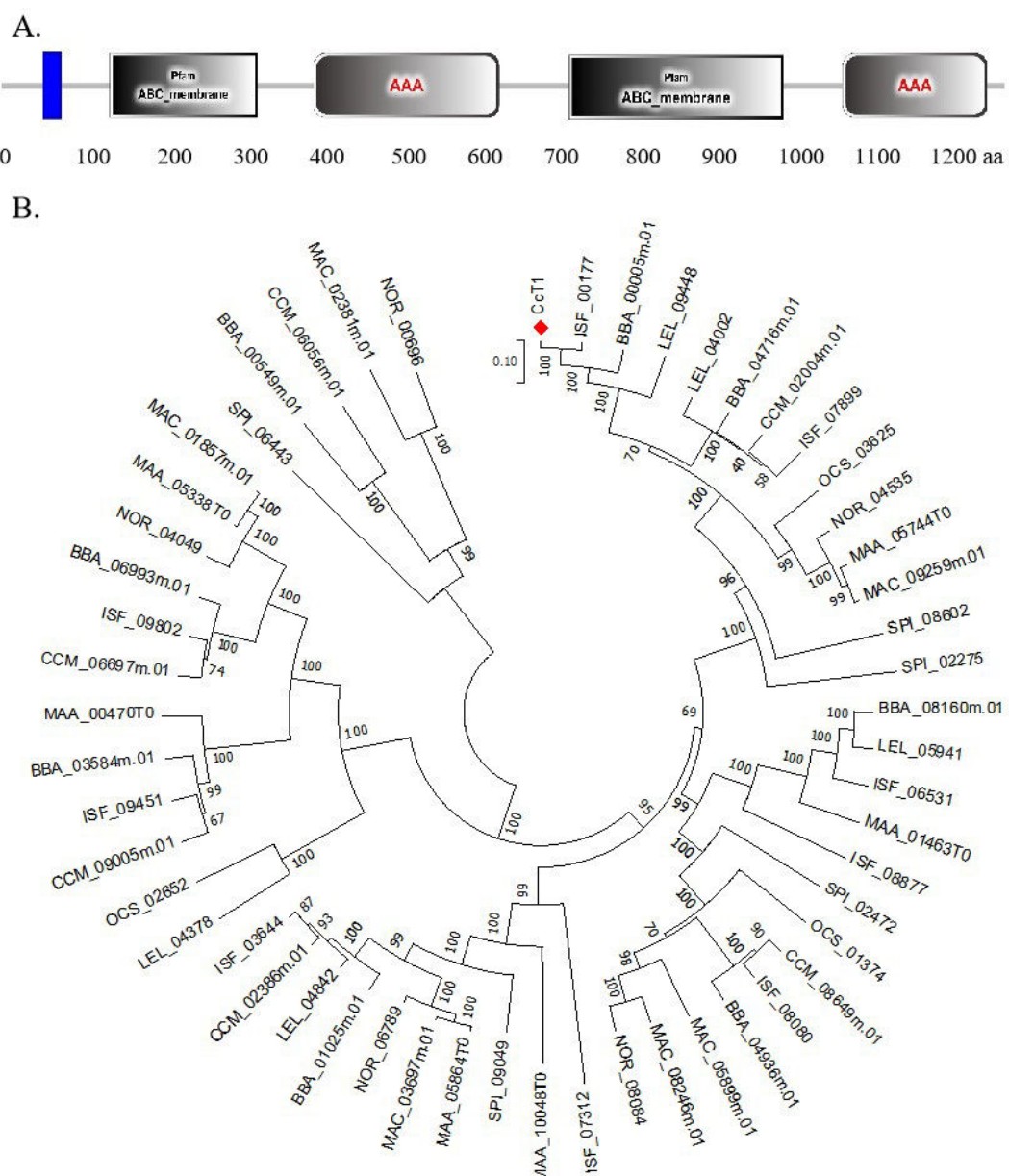

**FIG 2** Domain organization of CcT1 protein and the phylogenetic tree of *CcT1*. (A) Functional domains of ABC transporter CcT1 in *C. chanhua*. Blue rectangular area: transmembrane region; gray rectangular area: ABC_membrane domain; gray rounded rectangle area: AAA domain, ATPases associated with a variety of cellular activities. The number under the protein indicates the position of the domain and the protein length. The original sequence of *CcT1* is detailed in the SUPPLEMENTARY MATERIALS. (B) Phylogenetic tree of ABC transporter protein homologous proteins from entomogenous fungi. The amino acid sequences of ABC transporter protein homologous proteins from different species were downloaded from the NCBI database for phylogenetic analyses. The topology of this tree was generated using the neighbor-joining (NJ) method of MEGA7 with 2,000 bootstrap replicates. These numbers represent the percentage of replication trees (2,000 replications) with related taxa clustered together in the boot test.

Additionally, conidial yields were assessed at 7 and 14 days across different media (Fig. 6). The ΔCcT1 strain produced significantly fewer conidia than the WT on SDAY and 1/4 SDAY media at both time points (*P* < 0.05). On PDA media, no significant difference was observed at 7 days, but a notable reduction in conidiation was evident at 14 days in ΔCcT1 compared to the WT.

Furthermore, synnema formation was impaired in the ΔCcT1 strain. The result of the assay for synnema formation in the silkworm pupae showed that ΔCcT1 exhibited

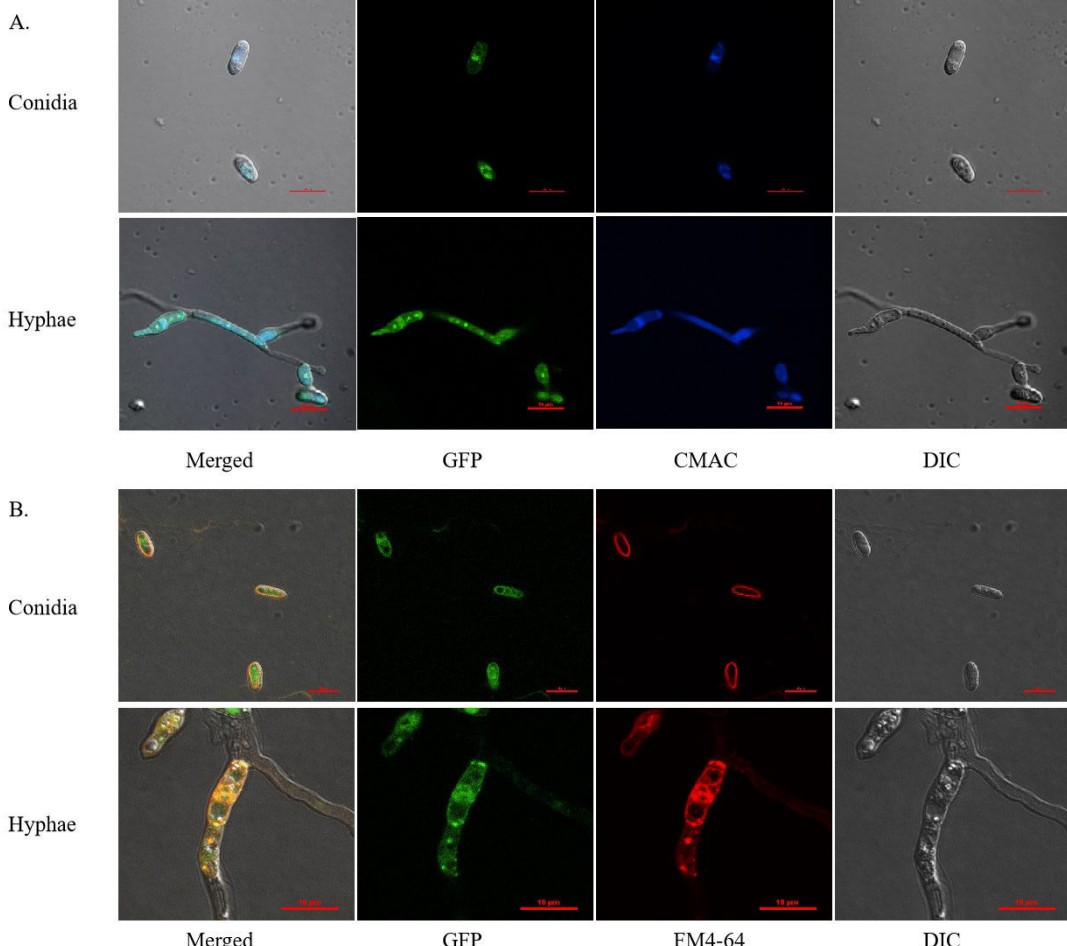

**FIG 3** Subcellular localization of *CcT1* in *C. chanhua*. The localization of *CcT1* in hyphae and conidia, showing that *CcT1* is located on the vacuole and cell membranes with the GFP fluorescent signal (expressed green). (A) All cells collected from the SDB culture at indicated time points were stained with the vacuole-specific dye CMAC (stained color in Klein blue). (B) All cells collected from the SDB culture at indicated time points were stained with the membrane-specific dye FM4-64 (stained color in red). Scale bars: 10 µm. DIC: differential interferometry.

reduced synnema production, whereas the complementation strain failed to restore this phenotype (Fig. 7). These results suggest that *CcT1* is essential for normal conidiation and synnema development in *C. chanhua*.

## Role of *CcT1* in oxidative stress response and cell wall integrity

To investigate the role of *CcT1* in the stress response, we exposed the WT and mutant strains to various chemical stressors, including menadione, hydrogen peroxide ($H_2O_2$), and Congo Red, by supplementing SDAY media. The ΔCcT1 strain displayed enhanced mycelial growth compared to the WT and C-ΔCcT1 strains on all tested plates containing oxidative agents (Fig. 8). Notably, on SDAY plates supplemented with 3 mmol/L $H_2O_2$, ΔCcT1 showed a 100% reduction in the inhibition ratio compared to the WT, indicating significantly increased resistance to oxidative stress. These findings suggest that *CcT1* negatively regulates antioxidant capacity and cell wall integrity, enhancing susceptibility to oxidative and cell wall-targeting agents when functional.

## Impact of *CcT1* on host cuticle penetration and fungal infectivity

To assess the role of *CcT1* in the host interaction, we performed assays to evaluate the penetration ability and infectivity of the WT and ΔCcT1 strains. Penetration assays

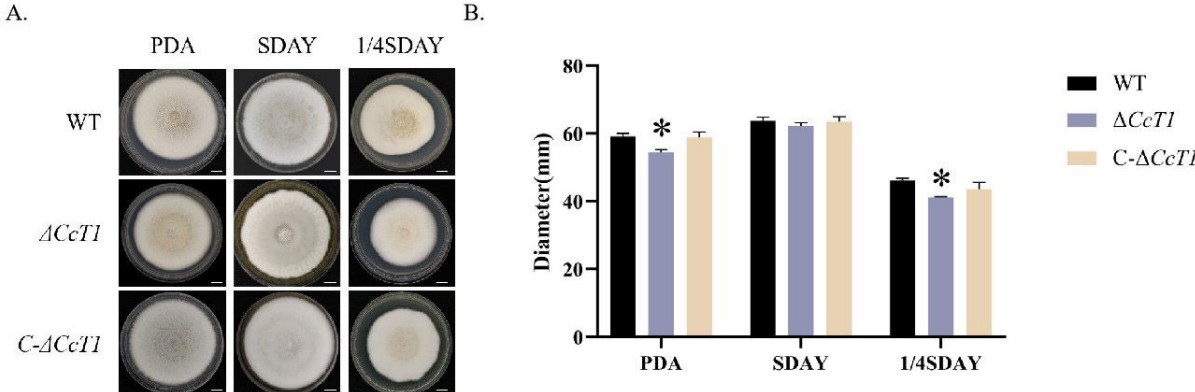

**FIG 4** Effects of *CcT1* deletion on the growth of *C. chanhua*. (A) Colony morphology of three strains growing on different media. Scale bars: 1 cm. (B) Colony diameters of three strains on PDA, SDAY, and 1/4SDAY media. The error bars represent the standard deviation (SD) of three biological replicates. *$P < 0.05$.

using cicada wings demonstrated that the ΔCcT1 strain exhibited a significantly reduced colony diameter (2.46 ± 0.07 cm) at the 4th day compared to the WT (2.84 ± 0.04 cm) (Fig. 9), indicating impaired ability to penetrate host cuticles.

However, infectivity assays involving intrahemocoel injection of *G. mellonella* larvae revealed no significant difference in the median lethal time ($LT_{50}$) among the WT (74.81 ± 0.13 hours), ΔCcT1 (72.53 ± 0.21 hours), and C-ΔCcT1 (74.18 ± 0.05 hours) strains (Fig. 10). These findings indicate that while CcT1 plays a crucial role in efficient cuticle penetration, its deletion does not substantially impact overall fungal infectivity once the cuticle barrier has been bypassed.

## *CcT1* is involved in BEA biosynthesis in *C. chanhua*

To determine the role of *CcT1* in BEA production, we quantified BEA levels in the WT and ΔCcT1 strains cultured on SDAY plates using high-performance liquid chromatography (HPLC). The ΔCcT1 strain exhibited a significant reduction in BEA content compared to the WT (Fig. 11), indicating that *CcT1* positively influences BEA biosynthesis.

Further analysis of gene expression via RT-qPCR revealed that the deletion of *CcT1* resulted in the significant downregulation of several key genes involved in the BEA biosynthetic pathway, including KIVR, phosphoglycerate mutase, pyruvate kinase, branched-chain amino acid aminotransferase (BCAT), 2-aminoadipate transaminase,

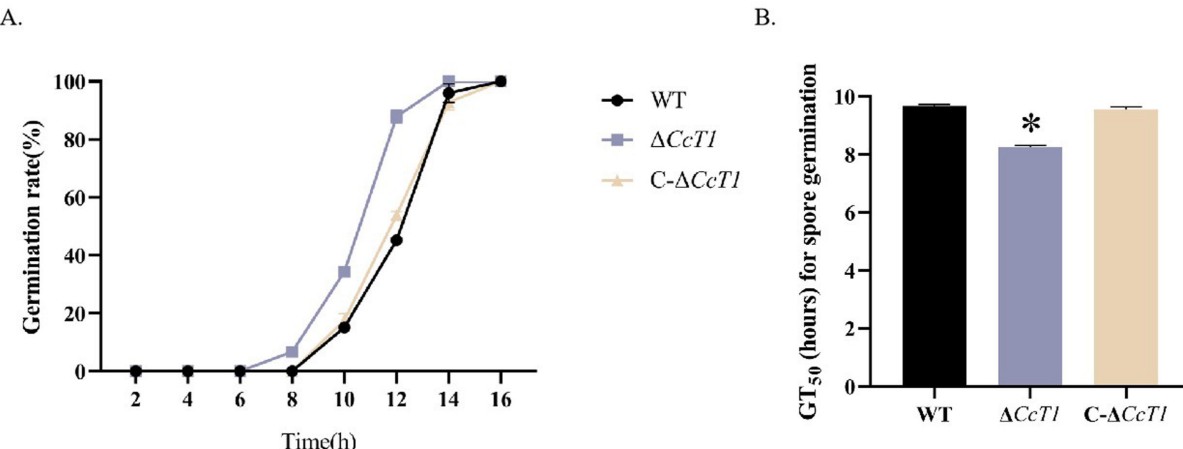

**FIG 5** Effects of *CcT1* deletion on conidial germination of *C. chanhua*. (A) The kinetics of conidial germination were evaluated by observing alterations in the germination rates of distinct strains at various time intervals following inoculation. (B) The half-time of germination ($GT_{50}$) for the respective strains was determined after they were cultured for 24 hours at 25°C. The error bars represent the standard deviation (SD) of three biological replicates. *$P < 0.05$.

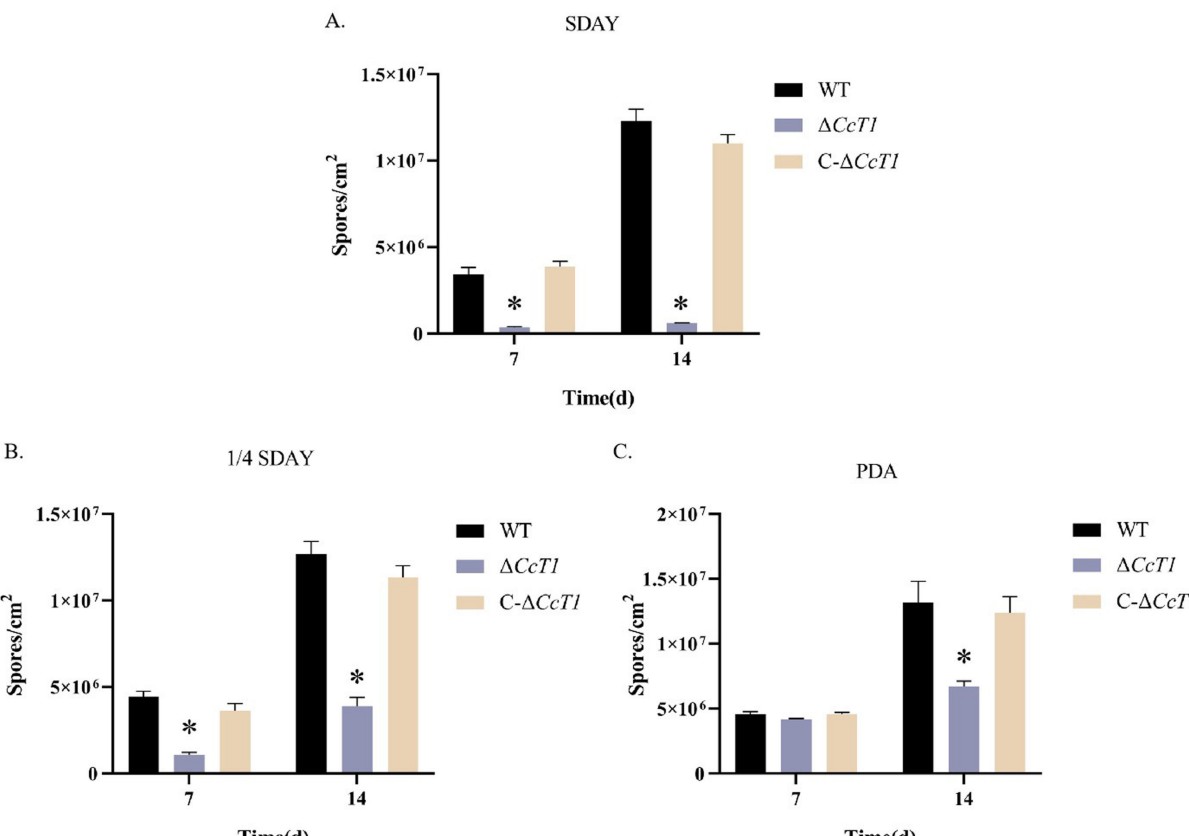

**FIG 6** Effects of *CcT1* deletion on the conidial yield of *C. chanhua*. Conidial yield of respective strains incubated for 7 and 14 days on different media. The error bars represent the standard deviation (SD) of three biological replicates. *$P < 0.05$.

fructose diphosphate aldolase, and glycine hydroxymethyltransferase (Fig. 11C). Conversely, the expression levels of the BEA synthetase genes *bea1* and *bea2* were upregulated, while those of *bea3* were downregulated in the Δ*CcT1* strain. These transcriptional changes suggest that CcT1 modulates BEA biosynthesis by regulating the expression of genes involved in precursor supply and enzymatic steps of the BEA synthesis pathway, ultimately leading to decreased BEA production upon CcT1 disruption.

## DISCUSSION

In this study, we characterized the ABC transporter CcT1 in *C. chanhua*, identifying its pivotal role in BEA biosynthesis, conidiation, oxidative stress resistance, and host cuticle penetration. The gene *CcT1* is strategically located within the BEA biosynthetic

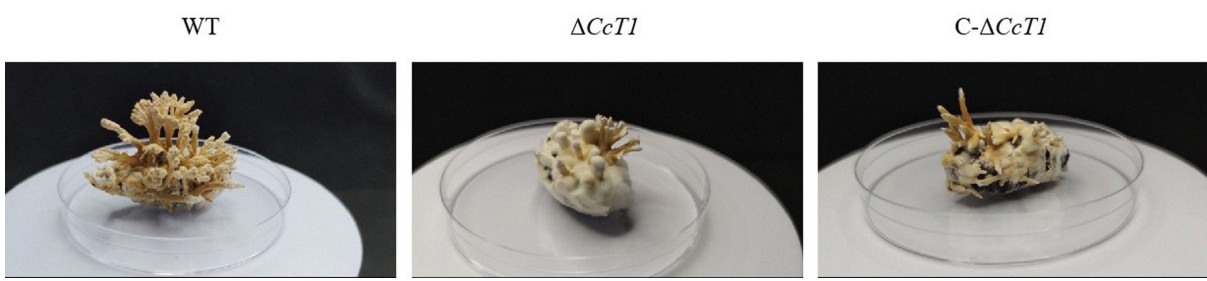

**FIG 7** Synnema production of *C. chanhua* on the pupae of Chinese tussah silkworm (*Antheraea pernyi*).

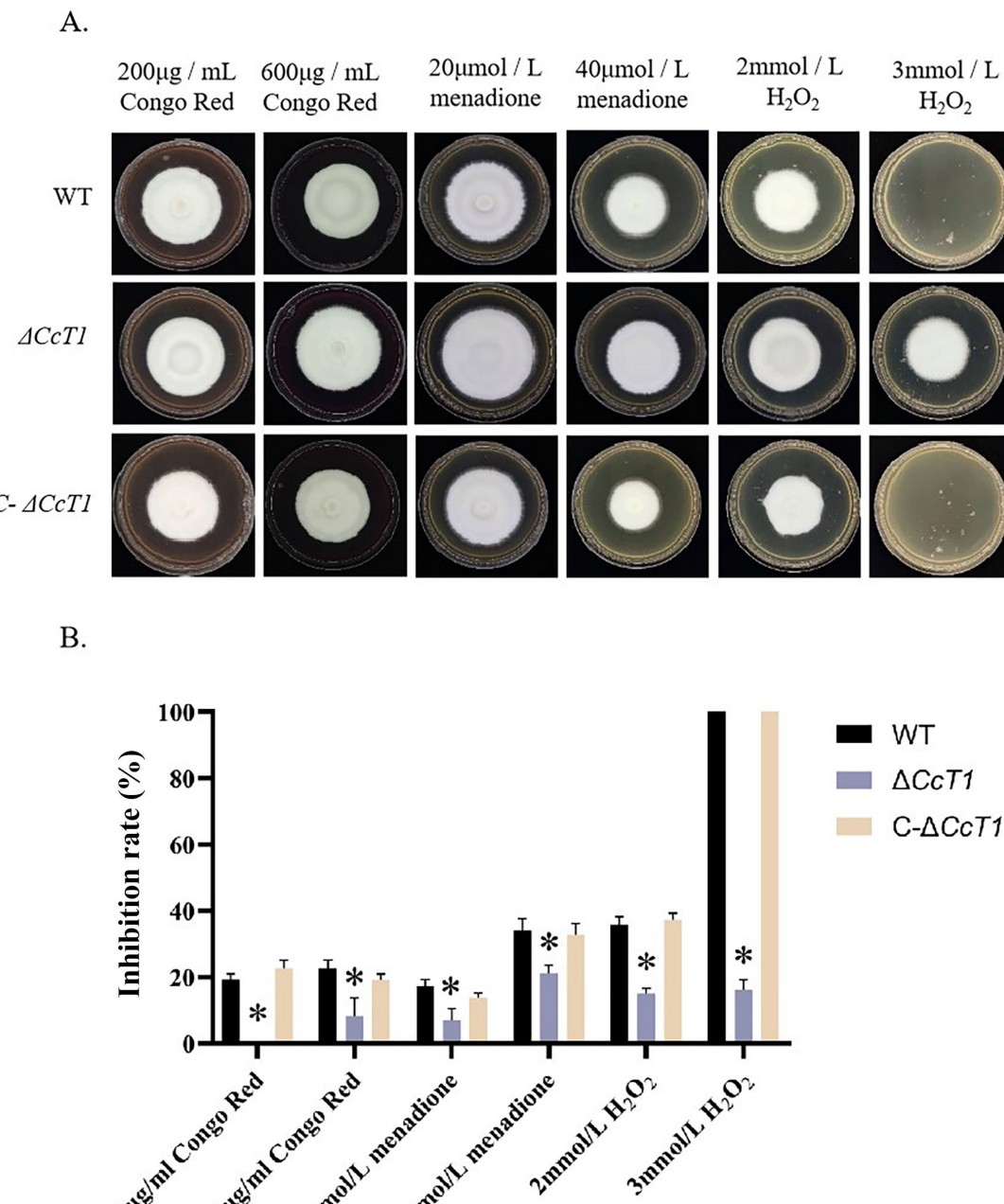

**FIG 8** Effect of *CcT1* deletion on stress resistance. (A) Colony morphology of three strains in SDAY containing different chemicals. (B) The relative inhibition rate of the strains cultured on the media containing Congo Red (200 µg/mL, 600 µg/mL), menadione (20 µmol/L, 40 µmol/L), or $H_2O_2$ (2 mmol/L, 3 mmol/L) after 14 days. The error bars represent the standard deviation (SD) of three biological replicates. *$P < 0.05$.

gene cluster, underscoring its potential regulatory influence on secondary metabolite production.

CcT1 was classified within the ABCB subfamily based on secondary structure and transmembrane domain predictions. The ABCB family is the largest ABC transporter subfamily and is prevalent across diverse fungal species, where members are involved in the export of pheromones, xenobiotics, and peptides and confer resistance to heavy metals (38). Previous investigations by Lu et al. (36), Zhang et al. (14), and Niehaus et al. (37) have identified ABC transporters within the BEA synthesis gene clusters of several BEA-producing fungi, including *Fusarium* spp., *C. fumosorosea*, and *C. chanhua*,

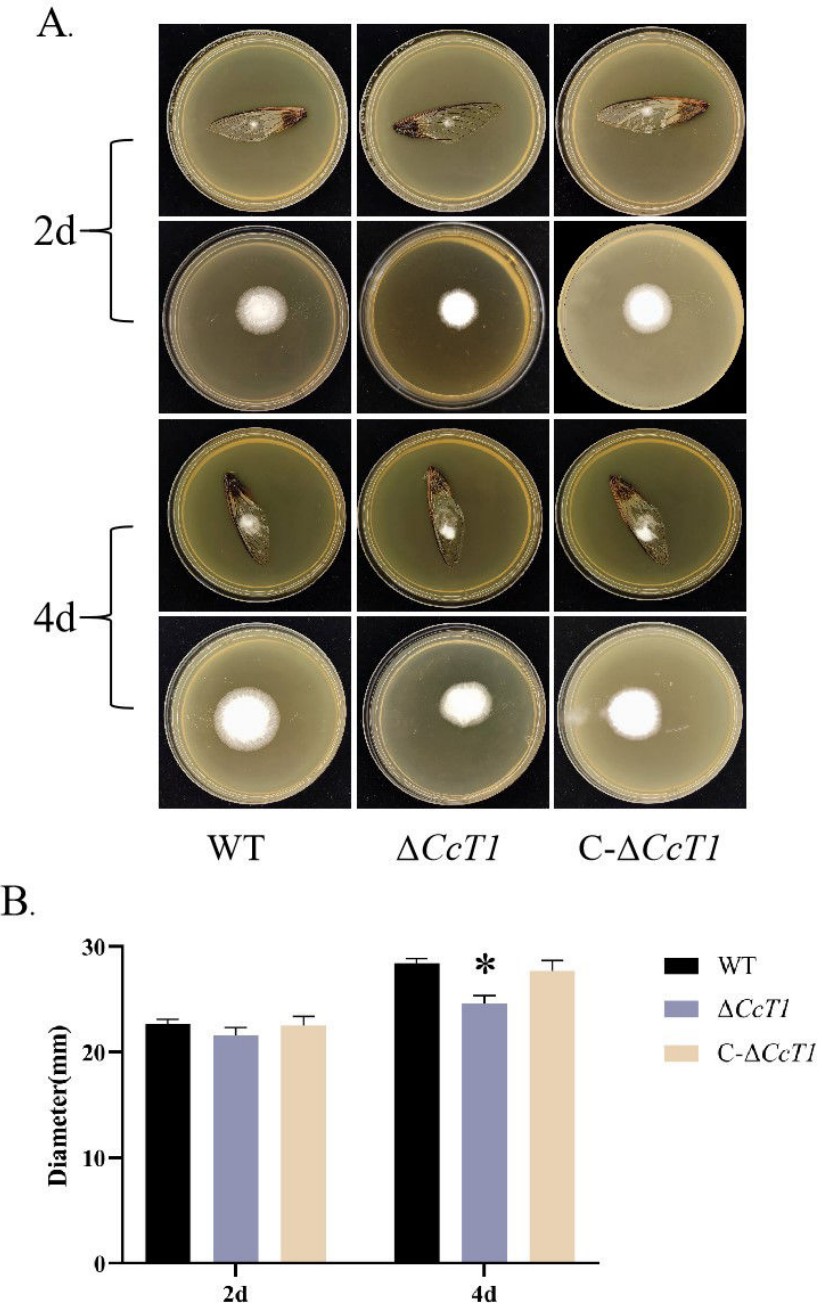

**FIG 9** Effect of *CcT1* deletion on its cuticle penetration ability. (A) After 2 or 4 days, cicada wings were removed, and fungal colony growth was measured following a 5-day cultivation at 25°C. (B) Colony diameters of respective strains incubated as shown in panel A. The error bars represent the standard deviation (SD) of three biological replicates. *$P < 0.05$.

but notably absent in *B. bassiana*. Zhang et al. (14) demonstrated that *F. proliferatum*, which harbors an ABC transporter gene within its *fpBeas* cluster, produces BEA at levels more than tenfold higher than *B. bassiana*. This observation suggests a contributory role of ABC transporters in enhancing BEA production in *Fusarium* species.

Contrary to the findings of Niehaus et al. (37), which indicated that the ABC transporter *Bea3* negatively regulates BEA synthesis in *F. fujikuroi*, our research demonstrated that disruption of *CcT1* in *C. chanhua* resulted in a substantial 64.22% reduction in BEA production compared to the wild-type strain. This discrepancy may stem from

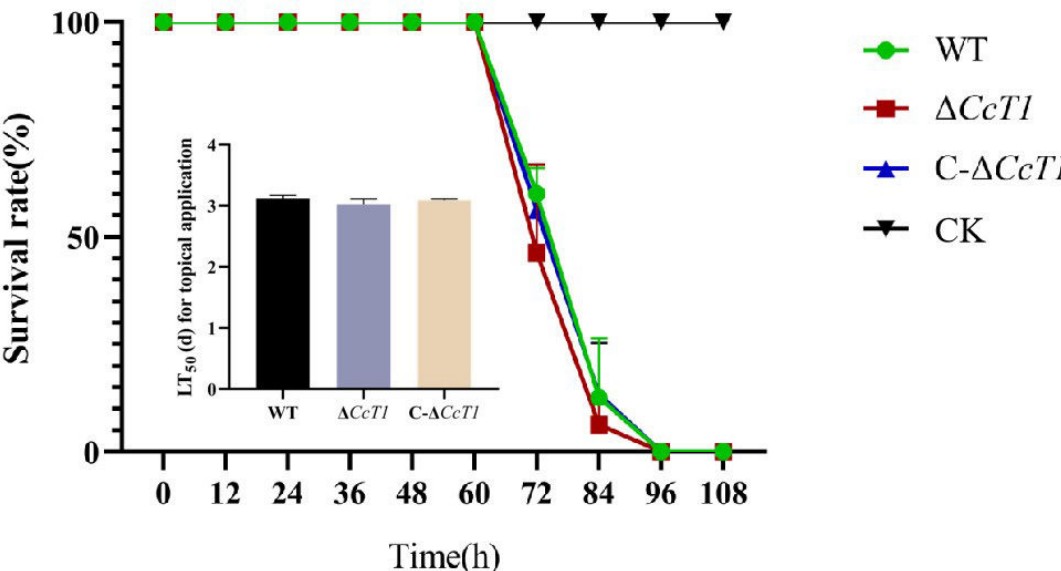

**FIG 10** Effect of *CcT1* deletion on fungal infectivity., survival rate, and LT50 (days) of *G. mellonella* after injection with conidial suspensions of three strains. The control insects were treated with Tween-80. The error bars represent the standard deviation (SD) of three biological replicates.

differences in the subcellular localization and functional dynamics of ABC transporters across different species. While *CcT1* is localized to the vacuole and cell membrane—potentially facilitating the storage and protection of BEA within cellular compartments—*Bea3* in *F. fujikuroi* is confined to the cytoplasmic membrane, primarily functioning in the export of BEA. The localization of *CcT1* in the vacuole and cell membrane suggests it may shuttle substrates between compartments, thereby supporting BEA biosynthesis to maintain high intracellular BEA levels. This mechanism promotes the stability of BEA and prevents its degradation, in contrast to Bea3, which appears to act more directly

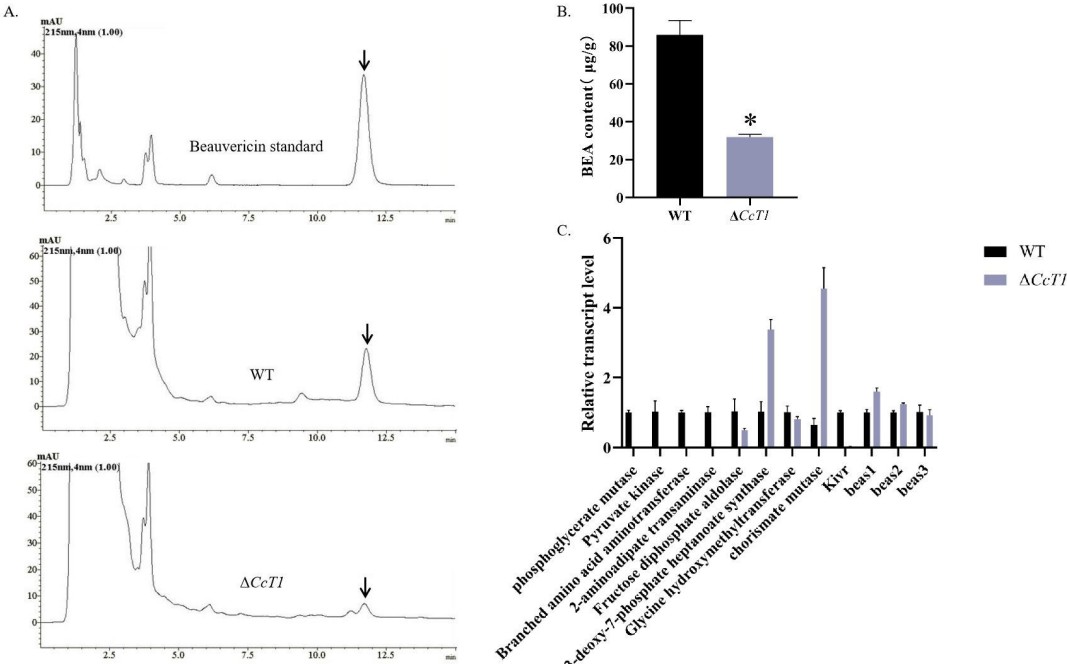

**FIG 11** Effect of *CcT1* deletion on the production of BEA. (A) HPLC analysis of BEA production in WT and ΔCcT1 of *C. chanhua*. (B) BEA content of WT and ΔCcT1; (C) expression levels of BEA synthesis-related genes. The error bars represent the standard deviation (SD) of three biological replicates. *$P < 0.05$.

in the efflux of toxins. Vacuoles are known to store metabolites such as branched-chain amino acids (BCAAs) and α-keto acids (α-KAs). CcT1 may export these metabolites into the cytosol, facilitating BEA biosynthesis. In the absence of *CcT1*, precursors may become trapped in vacuoles, leading to a depletion of resources in the cytosol and triggering the downregulation of biosynthetic genes.

RT-qPCR analysis revealed that *CcT1* disruption led to the downregulation of key BEA biosynthetic genes, including ketoisovalerate reductase (KIVR), phosphoglycerate mutase, pyruvate kinase, BCAT, 2-aminoadipate transaminase, fructose-bisphosphate aldolase, and glycine hydroxymethyltransferase. This downregulation implies *CcT1*'s regulatory role in BEA synthesis through precursor flux modulation, particularly affecting two critical metabolic nodes: glycolytic intermediates and amino acid metabolism.

BCAT catalyzes the transamination of branched-chain amino acids (BCAAs: e.g., valine, leucine, and isoleucine) to their corresponding α-KAs, which serve as substrates for both branched-chain fatty acid biosynthesis and secondary metabolite production. KIVR reduces ketoisovalerate to ketoisocaproate, which is a precursor for hydroxyisovaleric acid, a key component in BEA synthesis. Our findings suggest *CcT1* may facilitate BCAAs/α-KA transport (e.g., ketoisovalerate), with these metabolites exhibiting dual functionality: direct incorporation into BEA biosynthesis and modulation of nutrient-responsive signaling cascades. BCAAs and α-KAs may activate nutrient-sensing pathways such as the Target of Rapamycin (TOR) kinase, which regulates growth and secondary metabolism (39). In the *CcT1* knockout, the reduced BCAAs/α-KAs levels would suppress TOR signaling, leading to the downregulation of biosynthetic genes (e.g., BCAT and KIVR) as a means to conserve resources.

Interestingly, the expression levels of aromatic amino acid biosynthetic genes, specifically 3-deoxy-7-phosphoheptulonate synthase and chorismate mutase, were found to be upregulated in the Δ*CcT1* strain. This suggests a compensatory response aimed at maintaining aromatic amino acid homeostasis. However, despite the increase in the synthesis of precursors like L-phenylalanine, BEA production was compromised, highlighting the multifaceted regulatory role of *CcT1* beyond precursor supply.

The upregulation of chorismate mutase may redirect the flux of phenylalanine toward the production of tyrosine-derived antioxidants such as melanin and phenolic compounds, thereby bypassing the BEA biosynthetic pathway. This metabolic reorganization may provide an explanation for the paradox observed in the Δ*CcT1* strain-enhanced oxidative stress resistance despite compromised BEA production, particularly noteworthy given our previous findings demonstrating $H_2O_2$-induced upregulation of BEA biosynthesis under oxidative stress conditions (15).

The observed tolerance in the Δ*CcT1* strain likely arises from redirected metabolic investments and/or the activation of compensatory stress-response mechanisms. A speculative mechanism could involve *CcT1*-mediated transport of antioxidants; its functional loss might consequently elevate intracellular oxygen species (ROS) levels through impaired antioxidant trafficking. Such a redox imbalance would activate stress-responsive transcription factors (40), which may simultaneously suppress BEA biosynthetic genes while upregulating detoxification pathways.

Oxidative stress may inhibit enzymes like pyruvate kinase through oxidation of cysteine residues, creating a feedback loop that exacerbates metabolic disruption in the *CcT1* mutant (41). Furthermore, specific substrates transported by CcT1 could directly interact with transcription factors or epigenetic regulators (42); for instance, α-KAs may function as signaling molecules to activate or inhibit global regulators, thereby regulating secondary metabolic gene clusters (43).

The metabolic cost of synthesizing BEA may divert resources such as amino acids away from stress-response pathways. The disruption of CcT1 likely halts BEA production by blocking precursor transport or feedback regulation, while simultaneously triggering stress-response pathways that upregulate antioxidant defenses. In other words, the reduced production of BEA reflects a disruption in biosynthesis, while the enhanced

tolerance arises from redirected metabolic investment and/or activation of compensatory stress-response mechanisms.

The Δ*CcT1* strain exhibited delayed vegetative growth under nutrient-limited conditions (PDA, 1/4 SDAY) but showed wild-type growth on nutrient-rich SDAY medium, suggesting that CcT1 facilitates the transport of critical carbon or nitrogen sources (e.g., BCAAs or pyruvate) necessary for growth under stress. Notably, *CcT1* disruption accelerated conidial germination but reduced conidiation and impaired synnema formation. These phenotypic alterations correlate with transcriptional downregulation of glycolytic (such as pyruvate kinase) and BCAA metabolic genes (BCAT and KIVR), indicating a disrupted flux of key precursors essential for both energy metabolism and developmental processes. The downregulation of pyruvate kinase in Δ*CcT1* suggests disrupted carbon metabolism. CcT1 might transport pyruvate or regulate cytosolic ATP/ADP ratios through its ATPase activity. As an ATP-driven transporter, CcT1 could influence nuclear ATP pools, affecting chromatin-modifying enzymes that regulate gene expression. Low pyruvate or ATP levels could activate AMP-activated protein kinase (AMPK), a sensor of energy stress, which represses secondary metabolism to prioritize cell survival (44). This positions CcT1 as a metabolic integrator that links nutrient acquisition to developmental transitions by maintaining intracellular precursor pools.

From the above analysis, we can speculate that CcT1's dual localization on vacuole and cell membranes positions it as a key transporter for substrates like BCAAs, α-KAs, and antioxidants. Its role in maintaining precursor availability, metabolic signaling, and maintaining redox balance (via antioxidant transport) and energy status (via glycolysis intermediates) directly underpins BEA biosynthesis. *CcT1*'s regulatory role in gene expression stems from its molecular transport capacity, which shapes metabolic and redox states that feed into the signaling network.

Furthermore, the Δ*CcT1* strain exhibited reduced cuticle penetration on cicada wings (with a colony diameter reduced by 35%) but retained wild-type virulence in *G. mellonella*. It suggests that CcT1 may facilitate early infection by transporting cofactors for cuticle-degrading enzymes (e.g., metal ions) (45) or maintaining turgor pressure necessary for appressorial penetration (46). This indicates that *CcT1* plays a specific role in the initial stages of host infection, particularly in overcoming the physical barrier of the host cuticle, but does not influence the pathogenic mechanisms, which likely rely on intracellular nutrient acquisition systems.

Collectively, our findings position *CcT1* as a multifaceted regulator in *C. chanhua*, influencing secondary metabolite biosynthesis, nutrient assimilation, reproductive processes, stress responses, and host interaction capabilities. The regulatory role of *CcT1* in gene expression emerges from its transport activity, which shapes metabolic and redox states that feed into signaling networks. By controlling substrate availability (such as BCAA, α-KAs, and antioxidants) and maintaining energy/redox balance, CcT1 indirectly modulates nutrient-sensing pathways (like TOR and AMPK) that regulate biosynthetic genes and stress-responsive transcription factors that prioritize survival over secondary metabolism, compartmentalized metabolite pools that act as precursors or signals. The contrasting roles of ABC transporters in different fungal species underscore the evolutionary diversification of this protein family in adapting to specific ecological niches and metabolic requirements.

## Conclusion

In summary, this study elucidates the critical role of the ABC transporter CcT1 in *C. chanhua*. *CcT1* significantly influences BEA biosynthesis by regulating key metabolic pathways and gene expression within the BEA synthesis cluster. Additionally, *CcT1* is essential for optimal vegetative growth under nutrient-limited conditions, normal conidiation, and synnema formation, as well as maintaining the ability to penetrate host cuticles. The Δ*CcT1* mutant's enhanced resistance to oxidative stress further highlights the transporter's role in cellular homeostasis and stress response mechanisms. These insights not only advance our understanding of the molecular biology governing

secondary metabolite production and fungal physiology in *C. chanhua* but also identify *CcT1* as a potential target for molecular breeding strategies aimed at reducing BEA content. Such interventions could enhance the safety and therapeutic efficacy of *C. chanhua* as a traditional medicinal fungus by minimizing the accumulation of toxic metabolites without compromising its beneficial bioactive profiles.

## ACKNOWLEDGMENTS

We are grateful to Li Qin for technical assistance. We thank Deshui Yu and Yijun Zhang for contributing to the experimental work.

This research was funded by the Anhui Provincial Natural Science Foundation under grants 1908085MC56 and 2022AH050904.

## AUTHOR AFFILIATIONS

[1]Anhui Provincial Key Laboratory of Biological Control, Engineering Research Center of Fungal Biotechnology, Ministry of Education, Anhui Agricultural University, Hefei, China
[2]Xuanjiapu Ancient Ginkgo Forest Scenic Area Management Center, Taixing, China
[3]Jieshou Agricultural Service Station, Fuyang, China

## AUTHOR ORCIDs

Zhimin Liu http://orcid.org/0009-0006-9286-6275
Bo Huang https://orcid.org/0000-0001-6032-7396
Fan Peng http://orcid.org/0000-0002-5050-9572

## FUNDING

| Funder | Grant(s) | Author(s) |
| --- | --- | --- |
| Anhui Provincial Department of Science and Technology | 1908085MC56 | Fan Peng |
| Anhui Provincial Department of Education | 2022AH050904 | Fan Peng |

## ADDITIONAL FILES

The following material is available online.

### Supplemental Material

**Supplemental material (Spectrum03425-24-s0001.docx).** Fig. S1; Tables S1 and S2.

### Open Peer Review

**PEER REVIEW HISTORY (review-history.pdf).** An accounting of the reviewer comments and feedback.

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
