## [Reviewer comments · Microbiology Spectrum]

Microbiology Spectrum

A Putative ABC Transporter Gene, CcT1, is Involved in Beauvericin Synthesis, Conidiation and Oxidative Stress Resistance in *Cordyceps Chanhua*

Zhimin Liu, Jiahua Zhu, Ruixue Gong, Zhiyuan Wen, Yanwen Liu, Yulong Wang, Zengzhi Li, Bo Huang, and Fan Peng

Corresponding Author(s): Fan Peng, Anhui Agricultural University

Review Timeline:

Submission Date:	January 4, 2025
Editorial Decision:	January 25, 2025
Revision Received:	March 16, 2025
Editorial Decision:	March 27, 2025
Revision Received:	April 3, 2025
Accepted:	April 13, 2025

Editor: Chengshu Wang

Reviewer(s): Disclosure of reviewer identity is with reference to reviewer comments included in decision letter(s). The following individuals involved in review of your submission have agreed to reveal their identity: Yongjun Zhang (Reviewer #1); Caihong Dong (Reviewer #2)

Transaction Report:

DOI: <https://doi.org/10.1128/spectrum.03425-24>

Re: Spectrum03425-24 (A Putative ABC Transporter Gene, Cct1, is Involved in Beauvericin Synthesis, Conidiation and Oxidative Stress Resistance in Cordyceps Chanhua)

Dear Dr. Fan Peng:

Thank you for the privilege of reviewing your work. Below you will find my comments, instructions from the Spectrum editorial office, and the reviewer comments.

Revision Guidelines

Sincerely,
Chengshu Wang
Editor
Microbiology Spectrum

Reviewer #1 (Comments for the Author):

Liu et al. characterize a putative ABC transporter Cct1 located within the beauvericin (BEA) synthesis gene cluster in a traditional Chinese medicinal fungus Cordyceps chanhua. The authors report that disruption of Cct1 results in a decrease in BEA production, accompanying with downregulation of several BEA synthesis-related genes. They also found that Cct1 was involved in fungal growth, conidiation and germination, and oxidative stress response, and penetration of insect cuticle. These results

suggest that multiple roles of Cct1 in the fungus. Overall, this is an interesting study, but some crucial issues should be addressed.

Major Issues:

1. Importance section is missed.
2. Previous studies have reported a 13-fold increase in BEA production under oxidative stress, suggesting a positive correlation between BEA biosynthesis and the fungal tolerance oxidative stress in *C. chanhua* (<https://doi.org/10.3390/jof8050484>). In this study, the disruption of Cct1 significantly reduced BEA biosynthesis but paradoxically enhanced oxidative stress tolerance. The authors should provide an explanation for this issue.
3. A schematic diagram of the BEA biosynthetic gene cluster in *C. chanhua*, along with the putative functions of the genes, should be provided to facilitate understanding and comparison with others fungi. The authors reported that Cct1 was located within the BEA biosynthetic gene cluster and that its deletion affected the expression of BEA3. According to previous study in *Fusarium fujikuroi* (<http://10.1111/1462-2920.13576>), BEA3 encodes a transporter protein within the gene cluster. The authors should clarify the relationship between BEA3 and Cct1.
4. If Cct1 encodes a transporter protein within the BEA biosynthetic gene cluster, its primary function should involve the transport of related substances, rather than acting as a regulatory factor. Therefore, the use of the term "regulate" (line 310) is inappropriate. In the discussion, the authors should give the potential substrates of this transporter protein and its relationship with BEA biosynthesis, based on its localization and the phenotypes of the mutant strains.
5. The dye AMC used for subcellular localization should be clarified. Without co-localization results showing overlap between membrane staining and Cct1, the conclusion that Cct1 is localized to the cytoplasm membrane is not well-supported.

Minor Issues:

1. I suggest to delete the sentences 'ABC transporters are critical players in fungal metabolism, facilitating the transport of various substrates across cellular membranes. To elucidate the regulatory roles of CcT1, we conducted comparative analyses between the wild-type strain and a CcT1 knockout mutant' in the Abstract (line 20 to 23).
2. line 24, change 'RT-qPCR further' to 'RT-qPCR analysis'.
 1. Figure 1A, the specific meaning of the modules should be clearly explained in figure or the legend.
 2. Figure 3, there is an inconsistency between the colony size shown in the images and the quantitative data. For example, on PDA plates, there is no noticeable difference in colony size between the wild-type (WT) and the mutant strain, whereas the complemented strain appears significantly larger. On SDAY plates, the mutant strain shows markedly smaller colonies compared to the WT and complemented strains.
 3. Figure 7, the order of the images and bar charts should be consistent. Additionally, the concentrations of the chemical substances should be presented in the same order, either from low to high or high to low.
 4. Chromatograms from the HPLC analysis of BEA should be provided. If a standard compound was not used as a control, the authors should include mass spectrometry (MS) data to confirm that the detected metabolite is indeed BEA.
 5. Line 345, *F. proliferatum*, make sure it is *F. proliferatum* or *F. fujikuroi*?
 6. Line 25, changes 'half-time germination' to 'the time to reach 50% germination'.
 7. Line 305, lethal time 50, changes to "the median lethal time".
 8. Line 101, "*C. chanhua*" should be italic type.

Reviewer #2 (Comments for the Author):

The study addresses a critical topic concerning the regulation of mycotoxin production in *Cordyceps chanhua*, a fungus with significant medicinal value. The investigation of the ABC transporter gene CcT1 and its role in various biological processes (such as BEA synthesis, conidiation, and stress resistance) offers valuable insights into fungal metabolism and pathogenicity. The findings have important practical applications, particularly in reducing BEA levels through molecular breeding, which could enhance the safety and efficacy of *C. chanhua* as a medicinal agent.

Major Points:

1. The primary function of an ABC transporter is typically transport. Therefore, why does CcT1 play a regulatory role in gene expression? Expanding the discussion to explore the potential mechanisms underlying the observed effects of CcT1 deletion on BEA biosynthesis and stress resistance would be valuable. For instance, consider how the altered expression of specific genes (e.g., *kiv*, pyruvate kinase) might influence metabolic pathways and contribute to these observed phenotypes.
2. BEA production in *C. chanhua* increases under oxidative stress (Zhao et al., 2022). In contrast, this study shows that disruption of CcT1 enhances resistance to oxidative stress and significantly reduces BEA production compared to the wild-type strain. How can these findings be reconciled? A more detailed explanation of the relationship between stress resistance and BEA production in the context of CcT1 deletion is needed.
3. Line 234 states: "Using the complete genome sequence of *C. chanhua*, we identified the ABC transporter-encoding gene CcT1 by aligning it with the known ABC transporter protein from *C. fumosorosea* (ISF_00177) retrieved from the NCBI database." As the *C. chanhua* genome contains many ABC transporter genes, why was this particular gene chosen? If it was selected because it is located within the BEA synthesis gene cluster, why then was alignment performed with the ABC transporter protein from *C. fumosorosea*? Clarifying the rationale for this approach would improve the manuscript.
4. The statement that CcT1 localizes to both the vacuolar and cell membranes in hyphae and conidia requires further clarification. Please provide more detailed descriptions of the methodology used to determine this localization. Additionally, what does "AMC" refer to in Figure 2?

5. Transcriptomic Analysis vs. RT-PCR: Line 355 refers to "transcriptomic analysis"; however, the methods section only mentions quantitative real-time RT-PCR. Please clarify whether transcriptomic analysis was conducted or if RT-PCR was the primary technique used.

6. CLSM: Line 259 mentions CLSM. Please clarify this abbreviation

7. Reference Addition: Line 145 should include a reference to support the claim made.

Dear Editors and Reviewers:

Thank you for your review and valuable comments on our manuscript. Your feedback is very helpful in improving the quality of the paper. We're sorry for the late response, because we added a membrane staining experiment in subcellular localization to make our experimental results more reliable. We have carefully read all the comments and made corresponding revisions to the manuscript. Overall, we agree with and accept most of the reviewers' suggestions, and have made corresponding changes in the revised version. Additionally, we added an abbreviation list to make the expression clearer.

Below are the point-by-point responses and explanations of the revisions.

Reviewer #1:

1. Importance section is missed.

Thank you for your reminding, we added it in the manuscript.

2. Previous studies have reported a 13-fold increase in BEA production under oxidative stress, suggesting a positive correlation between BEA biosynthesis and the fungal tolerance oxidative stress in *C. chanhua* (<https://doi.org/10.3390/jof8050484>). In this study, the disruption of *Cct1* significantly reduced BEA biosynthesis but paradoxically enhanced oxidative stress tolerance. The authors should provide an explanation for this issue.

Thank you for your suggestion. About this tissue, we think that disrupting *CcT1* likely halts beauvericin production by blocking precursor transport or feedback regulation, while simultaneously triggering stress-response pathways that upregulate antioxidant defenses. We added the content in the discussion section (Line387-413).

3. A schematic diagram of the BEA biosynthetic gene cluster in *C. chanhua*, along with the putative functions of the genes, should be provided to facilitate understanding and comparison with others fungi. The authors reported that *Cct1* was located within the BEA biosynthetic gene cluster and that its deletion affected the expression of BEA. According to previous study in *Fusarium fujikuroi* (<http://10.1111/1462-2920.13576>), BEA3 encodes a transporter protein within the gene cluster. The authors should clarify the relationship between BEA3 and *Cct1*.

Thank you for your suggestion. A schematic diagram of beauvericin biosynthetic gene cluster in different fungi was added (Fig.1). From the diagram, it can be seen *CcT1* is also within BEA synthetic gene cluster, upstream of *kivr* gene, similar to BEA3 in *Fusarium fujikuroi* but genetically different upstream. Meanwhile they have different subcellular localization and therefore function differently. BEA3 is localized to cytoplasmic membrane, while *CcT1* is located to vacuole and cell membranes. We discussed the difference in the discussion section (Line 346-362).

4. If *Cct1* encodes a transporter protein within the BEA biosynthetic gene cluster, its primary function should involve the transport of related substances, rather than acting as a regulatory factor. Therefore, the use of the term "regulate" (line 310) is inappropriate. In the discussion, the authors should give the potential substrates of this transporter protein and its relationship with BEA biosynthesis, based on its localization and the phenotypes of the mutant strains.

Thank you for your suggestion. We think that *CcT1*'s regulatory role in gene expression emerges from its transport activity, which shapes metabolic and redox states that feed into signaling networks. "regulate" is not very appropriate, we changed it with "involve in".

And we think *CcT1*'s dual localization on vacuole and cell membranes positions it as a key transporter for substrates like BCAAs, α -keto acids,

and antioxidants. Its role in maintaining precursor availability, metabolic signaling, and maintaining redox balance (via antioxidant transport) and energy status (via glycolysis intermediates) directly underpins beauvericin biosynthesis. In the discussion section we expanded the discussion from the three aspects (Line363-413, 419-436).

5. The dye AMC used for subcellular localization should be clarified. Without co-localization results showing overlap between membrane staining and Cct1, the conclusion that Cct1 is localized to the cytoplasm membrane is not well-supported.

AMC is a dye for staining vacuole, its full name is 7-Amino-4-chloromethylcoumarin. In order to consistent with the cited literature (Li et al. 2022, reference 27), we changed AMC with CMAC. About the issue of localization, the membrane staining experiment was supplemented. More detailed descriptions were added in the method section (Line142-153) and the results section (Line 260-267, Figure 3B). From the results, it can be determined that Cct1 is localized to the cell membrane.

Minor Issues:

1. I suggest to delete the sentences 'ABC transporters are critical players in fungal metabolism, facilitating the transport of various substrates across cellular membranes. To elucidate the regulatory roles of CcT1, we conducted comparative analyses between the wild-type strain and a CcT1 knockout mutant' in the Abstract (line 20 to 23).

We appreciate your advice and deleted it.

2. line 24, change 'RT-qPCR further' to 'RT-qPCR analysis'

We appreciate your advice and changed it.

3. Figure 1A, the specific meaning of the modules should be clearly explained in figure or the legend.

Thank you for your suggestion. We provided explanations for the modules in Figure 2A.

4. Figure 3, there is an inconsistency between the colony size shown in the images and the quantitative data. For example, on PDA plates, there is no noticeable difference in colony size between the wild-type (WT) and the mutant strain, whereas the complemented strain appears significantly larger. On SDAY plates, the mutant strain shows markedly smaller colonies compared to the WT and complemented strains.

Thank you for pointing out this issue. After checking our experiment records, we found that it is a problem of image cropping. We have made appropriate adjustments to the images and added scale bars for more accurate assessment (Figure 4 in revised version).

5. Figure 7, the order of the images and bar charts should be consistent. Additionally, the concentrations of the chemical substances should be presented in the same order, either from low to high or high to low.

Thank you for your suggestion. We have made the adjustments to ensure consistency (Figure 8 in revised version).

6. Chromatograms from the HPLC analysis of BEA should be provided. If a standard compound was not used as a control, the authors should include mass spectrometry (MS) data to confirm that the detected metabolite is indeed BEA.

Thank you for your suggestion. We added chromatograms from the HPLC analysis of BEA of WT, gene knockout strain and BEA standard (Figure 11A in revised version).

7. Line 345, *F. proliferatum*, make sure it is *F. proliferatum* or *F. fujikuroi*?

Yes, it is a clerical error, it should be *F. fujikuroi*. Thank you. We corrected it.

8. Line 305, lethal time 50, changes to "the median lethal time".

We appreciated your advice and changed it.

9. Line 101, "*C. chanhua*" should be italic type.

We appreciated your advice and revised it.

Reviewer #2 (Comments for the Author):

1. The primary function of an ABC transporter is typically transport. Therefore, why does CcT1 play a regulatory role in gene expression? Expanding the discussion to explore the potential mechanisms underlying the observed effects of CcT1 deletion on BEA biosynthesis and stress resistance would be valuable. For instance, consider how the altered expression of specific genes (e.g., *kivr*, pyruvate kinase) might influence metabolic pathways and contribute to these observed phenotypes.

Thank you for your suggestion. We think that *CcT1*'s regulatory role in gene expression emerges from its transport activity, which shapes metabolic and redox states that feed into signaling networks. We revised the discussion (Line 363-413).

2. BEA production in *C. chanhua* increases under oxidative stress (Zhao et al., 2022). In contrast, this study shows that disruption of CcT1 enhances resistance to oxidative stress and significantly reduces BEA production compared to the wild-type strain. How can these findings be reconciled? A more detailed explanation of the relationship between stress resistance and BEA production in the context of CcT1 deletion is needed.

Thank you for your suggestion. About this tissue, we think that disrupting *CcT1* likely halts beauvericin production by blocking precursor transport or feedback regulation, while simultaneously triggering stress-response pathways that upregulate antioxidant defenses. We added the content in the discussion section (Line387-413).

3.Line 234 states: "Using the complete genome sequence of *C. chanhua*, we identified the ABC transporter-encoding gene CcT1 by aligning it with the known ABC transporter protein from *C. fumosorosea* (ISF_00177) retrieved from the NCBI database." As the *C. chanhua* genome contains many ABC transporter genes, why was this particular gene chosen? If it was selected

because it is located within the BEA synthesis gene cluster, why then was alignment performed with the ABC transporter protein from *C. fumosorosea*? Clarifying the rationale for this approach would improve the manuscript.

Thank you for your suggestion. We revised the content in results section (Line 225-237) as follows to make the approach clearer and rationale.

In our previous studies, an analysis of transcriptomic data under oxidative stress revealed that the expression levels of some putative ABC transporters were influenced by oxidative stress, with CL3361.Contig1_All showing significantly different expression levels. we utilized SMART and TMHMM online analysis software to predict its secondary structure and transmembrane characteristics. Its structure is presumed to resemble the (TMD-NBD)₂ configuration of the ABCB transporter. Consequently, we compared all known full-length ABCB amino acid sequences in entomogenous fungi and constructed a phylogenetic tree. This analysis showed that CL3361.Contig1_All shares the closest homology with the ABCB transporter ISF_00177 from *C. fumosorosea* and shared 94.28% amino acid sequence similarity. Furthermore, this gene is located within the BEA synthetic gene cluster. Given the significant changes in BEA content observed in *C. chanhua* under oxidative stress in previous studies, we speculate that this transporter is related to BEA biosynthesis, leading us to identify it as a focus of our research and designate it as *CcT1*.

4. The statement that CcT1 localizes to both the vacuolar and cell membranes in hyphae and conidia requires further clarification. Please provide more detailed descriptions of the methodology used to determine this localization. Additionally, what does "AMC" refer to in Figure 2?

Thank you for your suggestion, more detailed descriptions were added in the method section (Line142-153) and the results section (Line 260-267).

AMC is a dye for staining vacuole, its full name is 7-Amino-4-chloromethylcoumarin. In order to consistent with the cited literature (Li et al. 2022, reference 27), we changed AMC with CMAC.

Additionally, we added a membrane staining experiment to confirm *CcT1*'s localization to cell membranes (Figure 3B).

5. Transcriptomic Analysis vs. RT-PCR: Line 355 (346) refers to "transcriptomic analysis"; however, the methods section only mentions quantitative real-time RT-PCR. Please clarify whether transcriptomic analysis was conducted or if RT-PCR was the primary technique used.

The expression is not appropriate, only RT-qPCR verification was done.

We revised it.

6. CLSM: Line 259 mentions CLSM. Please clarify this abbreviation

The full name of CLSM is confocal laser scanning microscopy, we added the full name in the revised version.

7. Reference Addition: Line 145 should include a reference to support the claim made.

We have added a reference to support the claim made. The new reference (Tong et al., 2020, reference 28) provides additional support and context for our observations.

Re: Spectrum03425-24R1 (A Putative ABC Transporter Gene, CcT1, is Involved in Beauvericin Synthesis, Conidiation and Oxidative Stress Resistance in Cordyceps Chanhua)

Dear Dr. Fan Peng:

Thank you for the privilege of reviewing your work. Below you will find my comments, instructions from the Spectrum editorial office, and the reviewer comments.

I agree with both Reviewers' comments, please make corrections before final recommendations.

Revision Guidelines

Sincerely,
Chengshu Wang
Editor
Microbiology Spectrum

Reviewer #1 (Comments for the Author):

All issues I concerned about have been addressed. The experiments and the data have also been added accordingly. I have no additional comments.

Reviewer #2 (Comments for the Author):

The manuscript has been revised in accordance with the reviewers' comments and has seen significant improvement. I agree that "regulate" should be changed to "involve in." Please revise the entire text. Additionally, I find the explanation regarding the contradiction between "the disruption of Cct1 significantly reduced BEA biosynthesis" and "stress tolerance" not very convincing. Line 251 : Phylogenetic analysis placed CcT1 in close relation to these transporters within the ABC family. please rewrite the sentence

352 transporter. spell mistaken

Dear Editors and Reviewers:

Thank you for your review and valuable comments on our manuscript. Your feedback is very helpful in improving the quality of the paper. We have carefully read the comments and made corresponding revisions to the manuscript.

Below are the point-by-point responses and explanations of the revisions.

Reviewer #1 (Comments for the Author):

All issues I concerned about have been addressed. The experiments and the data have also been added accordingly. I have no additional comments.

Thank your for your recognition of the revision.

Reviewer #2 (Comments for the Author):

The manuscript has been revised in accordance with the reviewers' comments and has seen significant improvement. I agree that "regulate" should be changed to "involve in." Please revise the entire text. Additionally, I find the explanation regarding the contradiction between "the disruption of Cct1 significantly reduced BEA biosynthesis" and "stress tolerance" not very convincing.

Thank you for your reminding, we checked the entire text to make sure that "regulate" that needs to be modified was changed to "involve in".

Regarding the issue of contradiction between comprised BEA biosynthesis and increased stress tolerance, we hypothesize that disruption of the ABC

transporter-encoding gene may induce a complex metabolic rewiring that extends beyond secondary metabolite production. However, based on the currently available experimental results, we can only present the tentative hypothesis outlined in the manuscript. We sincerely appreciate your insightful observation regarding this critical question. In subsequent studies, we plan to conduct systematic researches to verify the proposed hypothesis and provide a more comprehensive explanation.

Line251 : Phylogenetic analysis placed CcT1 in close relation to these transporters within the ABC family. please rewrite the sentence

Thank you for your suggestion. We rewrite the sentence as “Phylogenetic analysis showed that CcT1 is evolutionarily closely related to these transporters in the ABC family”.

352 tnsporter. spell mistaken

Thank you for pointing out this mistake. We corrected it.

Re: Spectrum03425-24R2 (A Putative ABC Transporter Gene, CcT1, is Involved in Beauvericin Synthesis, Conidiation and Oxidative Stress Resistance in *Cordyceps Chanhua*)

Dear Dr. Fan Peng:

Minor issues have been corrected and this revision is now acceptable.

Your manuscript has been accepted, and I am forwarding it to the ASM production staff for publication. Your paper will first be checked to make sure all elements meet the technical requirements. ASM staff will contact you if anything needs to be revised before copyediting and production can begin. Otherwise, you will be notified when your proofs are ready to be viewed.

Sincerely,
Chengshu Wang
Editor
Microbiology Spectrum